# Factors Associated with the Detection of Inappropriate Prescriptions in Older People: A Prospective Cohort

**DOI:** 10.3390/ijerph182111310

**Published:** 2021-10-28

**Authors:** Núria Molist-Brunet, Daniel Sevilla-Sánchez, Emma Puigoriol-Juvanteny, Mariona Espaulella-Ferrer, Jordi Amblàs-Novellas, Joan Espaulella-Panicot

**Affiliations:** 1Hospital Universitari de la Santa Creu de Vic, 08500 Vic, Spain; mespaulella@chv.cat (M.E.-F.); jamblas@chv.cat (J.A.-N.); jespaulella@hsc.chv.cat (J.E.-P.); 2Central Catalonia Chronicity Research Group (C3RG), Centre for Health and Social Care Research (CESS), Universitat de Vic. University of Vic-Central University of Catalonia (UVIC-UCC), 08500 Vic, Spain; danielsevillasanchez@gmail.com; 3Pharmacy Department, Parc Sanitari Pere Virgili, 08023 Barcelona, Spain; 4Epidemiology department. Hospital Universitari de Vic, 08500 Vic, Spain; epuigoriol@chv.cat; 5Tissue Repair and Regeneration Laboratory (TR2Lab), University of Vic-Central University of Catalonia (UVIC-UCC), Fundació Hospital Universitari de la Santa Creu de Vic, and Hospital Universitari de Vic, 08500 Vic, Spain; 6Chair of Palliative Care, University of Vic, 08500 Vic, Spain

**Keywords:** frailty, polypharmacy, inappropriate prescription, multimorbidity, medication review, goal-oriented care

## Abstract

(1) Background: Ageing is associated with complex and dynamic changes leading to multimorbidity and, therefore, polypharmacy. The main objectives were to study an older community-dwelling cohort, to detect inappropriate prescriptions (IP) applying the Patient-Centred Prescription model, and to evaluate the most associated factors. (2) Methods: This was a prospective, descriptive, and observational study conducted from June 2019 to October 2020 on patients ≥ 65 years with multimorbidity who lived in the community. Demographic, clinical and pharmacological data were assessed. Variables assessed were: degree of frailty, using the Frail-VIG index; therapeutical complexity and anticholinergic and sedative burden; and the number of chronic drugs to determine polypharmacy or excessive polypharmacy. Finally, a medication review was carried out through the application of the Patient-Centred Prescription model. We used univariate and multivariate regression to identify the factors associated with IP. (3) Results: We recruited 428 patients (66.6% women; mean age 85.5, SD 7.67). A total of 50.9% of them lived in a nursing home; the mean Barthel Index was 49.93 (SD 32.14), and 73.8% of patients suffered some degree of cognitive impairment. The prevalence of frailty was 92.5%. Up to 90% of patients had at least one IP. An increase in IP prevalence was detected when the Frail-VIG index increased (*p* < 0.05). With the multivariate model, the relationship of polypharmacy with IP detection stands out above all. (4) Conclusions: 90% of patients presented one IP or more, and this situation can be detected through the PCP model. Factors with higher association with IP were frailty and polypharmacy.

## 1. Introduction

High-income countries face significant population ageing [1,2], which is associated with complex and dynamic changes that lead to the appearance of one or more chronic diseases, giving rise to multimorbidity [3]. Older patients with multimorbidity often meet frailty criteria [3].

Frailty is defined as an increased vulnerability to stressors resulting from a decrease in the physiological reserves of different systems [4]. It has been determined by identifying a critical number of impairments in physical strength, physical activity, nutrition, and mobility [4]. It is known that frailty is associated with a higher need for healthcare resources and predicts negative health outcomes such as appearance or worsening functional limitations, falls, hospitalisations, and mortality [5].

Epidemiological studies have associated multimorbidity and frailty with higher exposure to polypharmacy and the use of anticholinergic and sedative drugs. Polypharmacy is considered when the patient takes five or more medications continuously [6], and severe polypharmacy is when the number of chronic medications is ≥10 [7]. These facts have been associated with poorer outcomes, such as impaired cognitive and physical function, falls and hip fractures [8]. Furthermore, frailty and polypharmacy increase the risk of receiving inappropriate prescriptions (IP) [8,9,10,11]. This fact increases the risk of suffering adverse drug events (ADE) related to pharmacokinetics and pharmacodynamics changes by drug-drug interaction (DDI) associated with their multiple comorbidities [5,8,9,10,12,13]. Current evidence suggests that medication is often inappropriate in older patients [3,14], especially among frail individuals with polypharmacy and with a bad health self-assessment and comorbidities [11]

The use of several medications is the most documented independent risk factor to develop ADE, such as DDI, hospitalizations, cognitive and functional impairment, mortality, and healthcare expenditures, either in the overall population or, especially, in the older population [12,14]. Additionally, it is essential to remark that polypharmacy is also a risk factor for inappropriate pharmacological treatment adherence [15].

According to individual evolution, both concepts, appropriate and inappropriate prescription, are dynamic; thus, medications that previously could have been considered appropriate can become inappropriate depending on the progression of a chronic condition or the appearance of a new diagnosis that implies a change in the patient’s primary care goal. Consequently, depending on the patient’s characteristics and particular context, any medication can be potentially inappropriate [16].

There is agreement that constant vigilance and review are required when prescribing for these patients, considering the impact of every medication, the overall drug load, the presence of comorbidities, and function and care goals [8]. Pharmacological prescription in older patients has become a global concern because of a progressive, positive number of prescribed medications [6] and the increasing difficulties guaranteeing appropriate prescription to each patient profile [17].

Therefore, developing a specific tool to optimize prescription in older patients is crucial. This tool might consider the quality of life, functional status, main care goal, and life expectancy [18]. In this context, we propose a Patient-Centred Prescription (PCP) model as a methodology to optimize prescription in frail older patients. This approach combines clinical judgment and scientific evidence in a pragmatic and systematic process [19].

The objectives of the study were: (1) To determine the baseline situation and to calculate the frailty index (FI) of a cohort of older patients who lived in the community; (2) To assess the therapeutic plan through a PCP model and to analyse the prevalence of polypharmacy, number of IP, medication complexity and anticholinergic and sedative burden; and (3) To identify the variables that are potentially most related to IP.

## 2. Materials and Methods

### 2.1. Study Design and Subjects

This was a prospective, descriptive, and observational study on a cohort of older patients (from now on, the Community Older Patients cohort (COP cohort)) who lived in the community, either at home or in a nursing home. It was conducted from June 2019 to October 2020 in Osona, a semi-urban area in Catalonia (Spain).

Inclusion criteria: Patients 65 years of age or older, living in the community, either at home or in a nursing home with multimorbidity (two or more morbidities), that their primary care physician identified prescription management difficulties and requested a consultant team to review the pharmacological treatment.

Exclusion criteria: Patients who are probably living their last hours or days of life [20].

Ethics approval: We obtained verbal informed consent from patients or their main caregivers. Afterwards, we included the patient’s verbal informed consent in their electronic health record. The study was approved by the Scientific Ethics Committee, of each site: (1) FORES (Fundació d’Osona per la Recerca i l’Educació Sanitàries), under reference number 2019-106/PR237; (2) IDIAP Jordi Gol, under reference number 19/206-P; (3) Fundació Catalana d’Hospitals, under reference number CEI 20/23.

### 2.2. Data Collected

Personal data: Age and gender.

Functional data: Dependence or independence for medication management and the Barthel Index (BI) to assess basic activities of daily living were graded [21].

Medical data: we collected morbidities (from the diagnostic clusters within the Johns Hopkins University ACG system) [22] and adjusted-age Charlson Index [23]; dementia diagnosis, as stated in patients’ medical records, and the degree of deterioration established following the GDS (Global Deterioration Scale) [24]; blood pressure available in the last year; and geriatric syndromes.

Analytical data: Full blood count, sodium, potassium, urea, and glycosylated haemoglobin (HbA1c) were collected if available during the last year.

Pharmacological data: Number of chronic medicines prescribed for at least six months before the Medication Review (MR). It was determined if the patient had moderate polypharmacy (between 5 and 9 medications) or excessive polypharmacy (10 or more medications) [7]. Type of medication (qualitative classification) was recorded by ATC (Anatomical Therapeutic Chemical) system. Detection of therapeutical complexity through the MRCI [25] and DBI [26].

Frailty Index (FI): This variable was measured by the Frail-VIG index (“VIG” is the Spanish/Catalan acronym for Comprehensive Geriatric Assessment), which contains 22 simple questions that assess 25 different deficits [27,28]. FI was categorised as (1) no frailty (FI < 0.20); (2) mild frailty (FI 0.20–0.35); (3) moderate frailty (FI 0.36–0.50); and (4) severe frailty (FI > 0.50).

Patients in end-of-life (EOL) were identified according to the NECPAL CCOMS-ICO© tool criteria [29]. These patients are considered to be in the last months or the year of their life. The identification of EOL was based on: (a) the previous identification by the primary care team, (b) advanced disease criteria [29], or (c) Frail-VIG index >0.50.

Main therapeutic goal: According to the patients’ baseline situation, an individualized therapeutic goal was established: (i) survival when the patient’s baseline was optimal; (ii) functionality in patients in an intermediate situation; and (iii) symptomatic control in patients with a very vulnerable established baseline situation (patients in EOL situation were included).

### 2.3. Medication Review

Each patient’s pharmatherapeutic plan was reviewed through the application of the PCP model [19]. This model was a process with four systematic stages and a multi-disciplinary team carried it out made up of the patient’s primary care physician and nurse, with a consulting team (a geriatrician and a clinical pharmacist). The model focused all therapeutic decisions on the individualized global assessment of each patient: comprehensive geriatric assessment (CGA), the frailty index calculation (Frail-VIG index) [30], and the resulting individual therapeutic goal (prolonging survival maintaining functionality or prioritizing symptomatic control) [31]. The decisions were taken together with the patient or with their main caregiver in case of incapacity (Figure 1).

### 2.4. Inappropriate Prescription (IP)

With the MR, different criteria were used to determine IP; for example, in patients at EOL, Type 2 Diabetes Mellitus, hypertension and cardiovascular therapy, dyslipidaemia, mental health and dementia, pain, and osteoporosis.

Patients at EOL (NECPAL CCOMS-ICO© tool criteria [29]): according to STOPPFrail criteria, medications aimed at prolonging survival and those for primary prevention were assessed for potential discontinuation. Medications for secondary prevention were individualised based on patient goals [31,32].

Type 2 Diabetes Mellitus (T2DM): Two main proposals were used to individualise hypoglycaemic treatment: (a) therapeutic intensity criteria, following the American Diabetes Association (ADA) recommendations [33,34,35]; (b) type of medication: sulphonylureas (SU) were considered inappropriate because of the high risk of hypoglycaemia [34,36]; metformin was considered inappropriate if there were non-adjusted doses in cases of renal failure [34]; glifozins (SGLT2 inhibitors) were considered inappropriate when it was prescribed in patients without heart failure and chronic renal failure (glomerular filtration rate (GFR) < 45 mL/min) [34,37]; and short-acting insulin or mixtures were also considered inappropriate, except when it could be justified [34]. Table 1 describes the therapeutic goals in T2DM according to the patient profile.

Hypertension (HT) and Cardiovascular Therapy: There is currently evidence suggesting less intensive monitoring in people with multimorbidity, particularly in cases of dementia or limited life expectancy [38]. Globally, blood pressure under 140/90 mmHg has been associated with a higher risk of falls and mortality [39,40,41]. We considered an antihypertensive medication as an IP in EOL patients when the patient’s mean systolic blood pressure has been lower than 130 mmHg over the last year [31].

Dyslipidaemia: Statins are not recommended in EOL patients [32], regardless of the indication, particularly in primary prevention cases. In secondary prevention, we can individualise decision-making based on each patient’s associated risks and benefits [31]. We considered a lipid-lowering drug as an IP when prescribed to a patient with a total cholesterol level under 150 mg/dL, because it is a malnutrition marker [41].

Mental Health and Dementia: The European Association of Palliative Care’s recommendations were used to make decisions; they propose a different therapeutic main goal in patients with dementia according to the stage of their pathology, based on evidence and consensus among experts [42]. We considered chronic antipsychotic drugs as an IP when prescribed to patients without behavioural disorders over the last 3–6 months or when prescribed to treat insomnia, as there is no evidence to support this indication [31,42,43].

Pain: Following Beers/STOPP criteria, the following proposals were made [36,44,45,46]: (a) Tricyclic antidepressants were considered an IP because of their anticholinergic effects; (b) non-steroidal anti-inflammatory drugs (NSAIDs) were considered inappropriate when they were not prescribed at the lowest dose or for the shortest time possible, because of their high risk of ADEs; (c) weak opioids such as tramadol and codeine were registered as IP unless prescribed at low doses, due to the risk of ADEs; (d) major opioids, such as morphine and oxycodone, were considered IP if they were not associated with a laxative; and (e) meperidine was considered an IP due to its anticholinergic potential.

Osteoporosis: We considered calcium supplements (except in cases of symptomatic hypocalcaemia), vitamin D, or anti-resorption drugs as inappropriate in EOL patients [32].

Other groups: Based on the PCP model, the medications that could not be considered a correct indication or the most optimised posology were recorded as IP.

### 2.5. Sample Size

IP prevalence in the frail older population was estimated at 71% to calculate sample size [47]. With a 95% confidence level and 5% accuracy, a minimum of 352 patients should be included.

### 2.6. Statistical Methods

IBM SPSS Statistics v27.0 statistical software was used to perform statistical analysis. The results for categorical variables were described as absolute and relative frequencies. Outcomes for continuous variables were expressed by means and standard deviations (SD). The statistical tests used to evaluate the relationship between two qualitative variables were the Chi-square test (or Fisher’s exact test in 2 × 2 tables where the expected frequencies were <5). The Student’s *t*-test was used to analyse the relationship between quantitative and qualitative variables. To identify the factors associated with IP, we used univariate and multivariate logistic regression. Statistical significance was established when the value of *p* was under 0.05.

## 3. Results

### 3.1. Subject Baseline Data

A total of 428 patients were enrolled (66.6% women). The mean age was 85.5 years (SD 7.67). Almost half of them lived in a nursing home (50.9%). Globally, they had moderate dependence for basic daily activities, with a mean Barthel Index of 49.93 (SD 32.14), a prevalence of frailty of 92.5%, with 73.8% of patients suffering some degree of cognitive impairment. Table 2 outlines the COP-cohort’s baseline demographic, clinical, functional, and cognitive data, and Table 3 lists the baseline pharmacological data. Globally, a particularly high prevalence of IP was detected. Up to 90.0% of the patients had at least one IP.

Moreover, an increase in the prevalence of IP was detected when the Frail-VIG index increased (*p* < 0.05) (Figure 2).

### 3.2. Data of IPs

Table 4 shows the descriptive analysis of the baseline situation and the number of IPs. The clinical variables most associated with presenting at least one IP were BI and the number of morbidities. BI had a mean of 60.7 in patients without IP and 48.7 in patients with at least one IP (*p* = 0.020). Regarding the morbidities, 6.9% of patients without IP presented five or more morbidities, and the percentage was 93.1% when they had at least one IP (*p* = 0.011).

Regarding morbidities, 51.1% of patients without IP had HT and 69.6% of patients with at least one IP had HT (*p* = 0.003); 4.7% of patients without IP had T2DM and 21.8% of them had at least one IP had T2DM (*p* = 0.008). Concerning FI, there was also an association with IP. Patients with one or more IPs presented a higher mean of FI than those without IP (0.39 (SD 0.1) vs. 0.34 (SD 0.1) (*p* = 0.023)).

Figure 3 and Figure 4 show the prevalence of each polypharmacy degree and the prevalence of MRCI degree considering the number of IPs. In both, all comparisons showed statistically significant differences (*p* < 0.001). Remarkably, patients with no IPs presented the lowest polypharmacy and MRCI rates. Moreover, on the contrary, those patients with more IPs had increased polypharmacy and MRCI rates (*p* < 0.001). Figure 5 shows the prevalence of the DBI degree considering the number of IPs, and differences were statistically significant in all groups, except when IPs were 0 versus ≥1.

Table 5 outlines the types of IP analysed according to the Anatomical, Therapeutic, Chemical (ATC) classification. It is essential to highlight that the groups most frequently prescribed inappropriately were ATC A (alimentary tract and metabolism), B (blood and blood-forming organs), C (cardiovascular system), and N (nervous system), with a percentage of 24.72%, 9.29%, 30.85%, and 24.62%, respectively.

### 3.3. Univariate Analysis and Multivariate Analysis

Table 6 shows the univariate analysis, which highlights the relationship of the detection of IP with the following variables: T2DM, number of morbidities (especially if they were ≥5), the Frail-VIG index (severe frailty), polypharmacy (both moderate as well as excessive), therapeutic complexity (high complexity), and DBI (high DBI).

With the multivariate model, the relationship of polypharmacy with IP detection stands out above all, both moderate and excessive. The frailty index was a significant predictor factor in the univariate model for those with a high IPs presence (≥2 or, or ≥3), but this did not remain significant in the multivariate analysis.

## 4. Discussion

In this study describing a sample of older patients recruited at the community level, we detected a high prevalence of functional and cognitive impairment and frailty in a specific health region with semi-urban characteristics. Similarly, high rates of moderate and excessive polypharmacy, therapeutic complexity, and anticholinergic and sedative burden with the pharmacological data were observed. These results are higher than might be expected in a standard cohort with patients aged 65 or older [14,48]. This fact could be explained by the inclusion criteria that selected patients according to one objective criterion (presenting multimorbidity), as well as one subjective criterion (patients with multimorbidity whose primary care physician identified prescription management difficulties).

The application of the PCP model detected a prevalence of IPs of up to 90%. This result is a much higher proportion of IPs than those detected in other studies using explicit criteria (Beers and STOPP-START criteria) [49]. This data is probably due to two main reasons: (i) inclusion criteria with patients with multimorbidity [50,51] and (ii) the PCP model allows the optimisation of individualised medication, thus resulting in a more thorough analysis of the prescription.

Thus, PCP should generally be considered an advanced MR (based on medication history, patient information, and clinical information) that optimises the prescription process [52].

Regarding ATC groups, we found that four of the 13 groups included in this classification accounted for almost 90% of IPs (alimentary tract and metabolism, blood and blood-forming organs, and cardiovascular and nervous system). Once again, this shows that IP is usually concentrated in a small number of pharmacological groups [53,54]. According to the multivariate analysis, it is remarkable that polypharmacy is the variable most commonly associated with the IP presence.

Notably, the positive relationship between frailty and IP was detected in this descriptive study. Furthermore, in the univariate model, a relationship between the FI and IP was observed. Nevertheless, in the multivariate model, this relationship disappeared. This fact could be due to FI being the result of summarising all the other data analysed. Indeed, this study could help open a path towards new studies to investigate the relationship between frailty and IP.

The current study has some limitations, such as the lack of a larger sample of non-frail patients, which would allow us to conduct a more accurate statistical analysis.

As a future goal, it would be interesting to assess clinical and pharmacological outcomes after applying different proposals to individualise the therapeutic approach through a longitudinal follow-up study.

## 5. Conclusions

The application of the PCP model in older adults with multimorbidity enabled to identify up to 90% of them presenting at least one IP. Frailty had a positive association with IP detection, and polypharmacy was the most involved factor in IP detection.

However, more studies should be performed with frail and non-frail patients to validate the potential of this tool.

## Figures and Tables

**Figure 1 ijerph-18-11310-f001:**
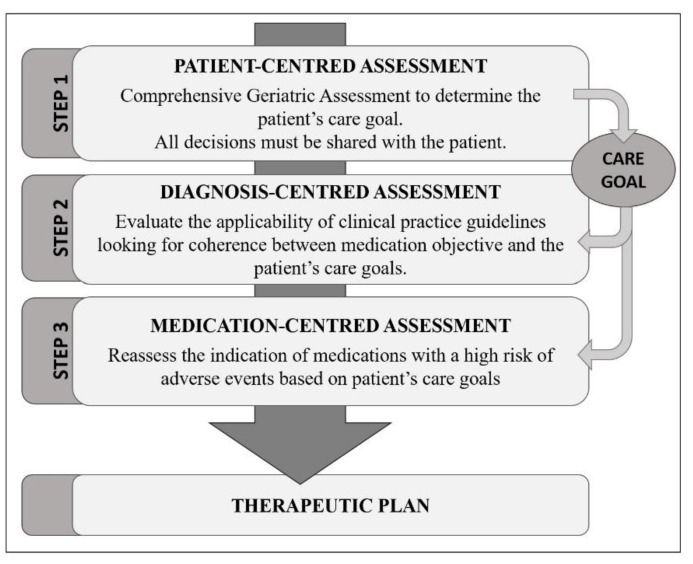
Patient-Centred Prescription model.

**Figure 2 ijerph-18-11310-f002:**
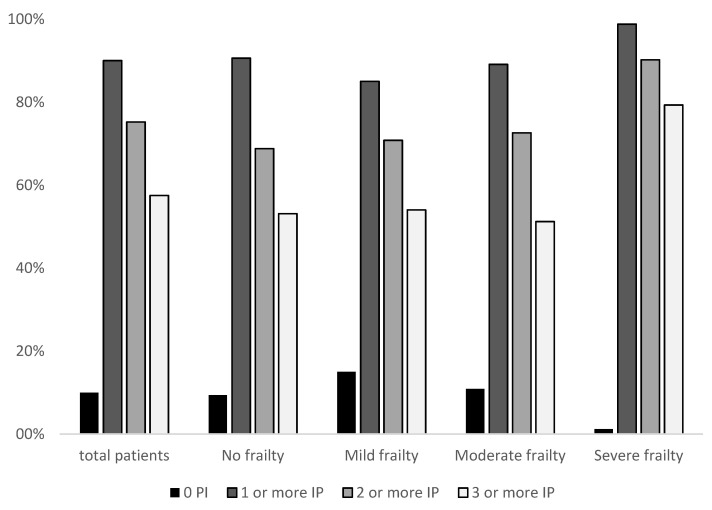
Number of inappropriate prescriptions (IP) according to the frailty index (FI).

**Figure 3 ijerph-18-11310-f003:**
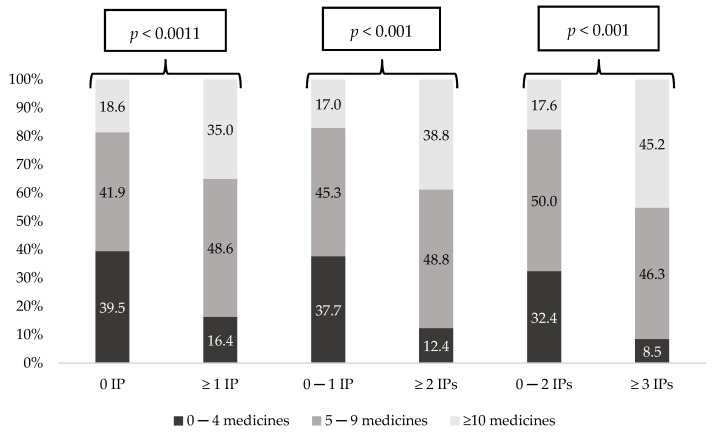
Prevalence of polypharmacy degree considering number of Inappropriate Prescriptions (IPs).

**Figure 4 ijerph-18-11310-f004:**
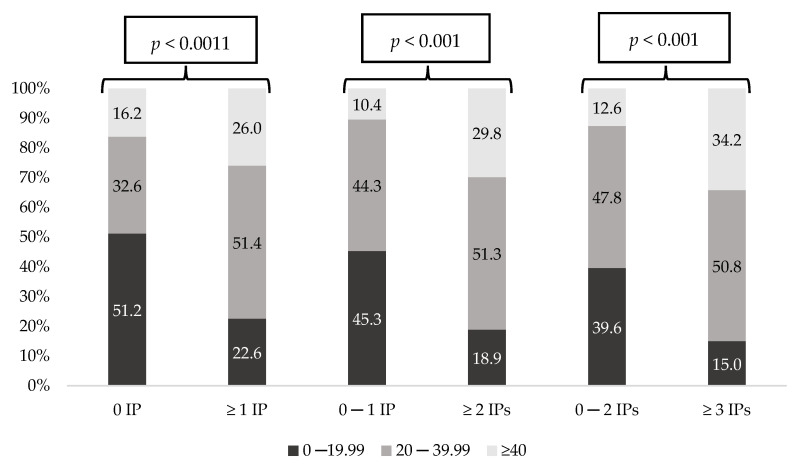
Prevalence of MRCI degree considering number of Inappropriate Prescriptions (IPs).

**Figure 5 ijerph-18-11310-f005:**
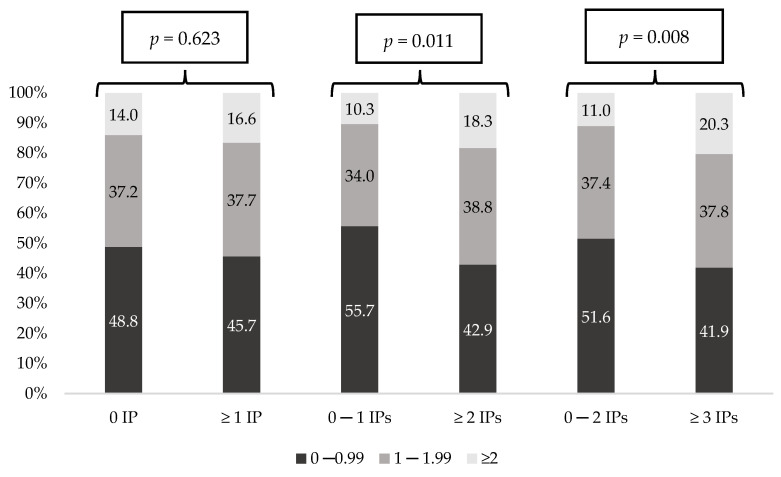
Prevalence of DBI degree considering the number of Inappropriate Prescriptions (IPs).

**Table 1 ijerph-18-11310-t001:** Type 2 Diabetes Mellitus (T2DM) therapeutic goals considering patient profile.

Target	Patients		
	Healthy Older Adults *	Frail Older Adults ^†^	Older Adults in a Probable EOL Situation ^‡^
Qualitative Glycaemic	Similar to those for diabetic young adults	Assess the decrease of therapeutic intensity	Quality of life preservation **
Quantitative Hba1c^¶^	≤7–7.5%	≤8.0%	Avoid reliance on A1C **
Therapeutic Goal ^††^	Prolong survival	Maintain functionality	Symptomatic treatment

* Good functional and cognitive status, and long life expectancy. ^†^ With functional disability and dementia or moderately limited life expectancy. ^‡^ End-of-life (EOL) situation, understood as a period of 1–2 years. Hba1c^¶^, glycated haemoglobin. ** Glucose control decisions should be based on avoiding hypoglycaemia and symptomatic hyperglycaemia episodes. ^††^ Based on the Patient Centred Prescription (PCP) Model.

**Table 2 ijerph-18-11310-t002:** COP-cohort’s baseline data.

Baseline Data		Total N = 428
Demographic Data		
Age, mean (SD)		85.52 (7.67)
Gender, N (%)	Men	143 (33.4%)
	Women	285 (66.6%)
Origin, N (%)	Home	210 (49.1%)
	Nursing Home	218 (50.9%)
Clinical, Functional and Cognitive Data		
Medication self-management *		58 (27.6%)
Barthel Index (BI), mean (SD)		49.93 (32.14)
BI (degrees)	Independence (BI ≥ 95)	51 (11.9%)
	Mild dependence (BI 90–65)	120 (28.0%)
	Moderate dependence (BI 60–25)	129 (30.2%)
	Severe dependence (BI ≤ 20)	128 (29.9%)
Cognitive status	No dementia	112 (26.2%)
	Mild dementia	62 (14.5%)
	Moderate dementia (GDS 5 to GDS 6B)	112 (26.2%)
	Advanced dementia (from GS 6C)	142 (33.1%)
Geriatric Syndromes (GS), mean (SD)		2.92 (1.52)
Type of GS	Falls	144 (33.6%)
	Dysphagia	84 (19.6%)
	Pain	99 (23.1%)
	Depressive syndrome	198 (46.3%)
	Insomnia	229 (53.5%)
Morbidities, mean (SD)		4.91 (2.16)
Morbidities (number)	1–2	43 (10.0%)
	3–4	168 (39.3%)
	5 or more	217 (50.7%)
Morbidities (type)	Hypertension	290 (67.8%)
	Chronic renal failure	186 (43.5%)
	Type 2 Diabetes	110 (25.7%)
	Heart Failure	88 (20.6%)
Charlson Index, mean (SD)		3.26 (2.27)
Frailty (FI), mean (SD)		0.39 (0.13)
FI (degrees)	No frailty (0–0.19)	32 (7.5%)
	Mild frailty (0.20–0.35)	113 (26.4%)
	Moderate frailty (0.36–0.50)	201 (47.0%)
	Severe frailty (0.51–1)	82 (19.1%)
End-of-life patients		155 (36.2%)
Therapeutic aim	Survival	41 (9.6%)
	Functionality	223 (52.1%)
	Symptomatic	164 (38.3%)

* Only patients living at home were assessed (N = 210).

**Table 3 ijerph-18-11310-t003:** Baseline pharmacological data for the COP-cohort.

Baseline Pharmacological Data		Total N = 428
Polypharmacy, mean (SD)		8.13 (3.88)
Polypharmacy (degree)	0–4 medications	80 (18.7%)
	5–9 medications	205 (47.9%)
	10 or more medications	143 (33.4%)
Medication Regimen Complexity Index (MRCI), mean (SD)		30.74 (16.26)
MRCI (degree)	Low complexity (0–19.99)	109 (25.5%)
	Moderate complexity (20–39.99)	208 (48.6%)
	High complexity (40 or more)	111 (25.9%)
Drug Burden Index (DBI), mean (SD)		1.17 (0.84)
DBI (degree)	Low DBI (0–0.99)	70 (16.4%)
	Moderate DBI (1–1.99)	197 (46.0%)
	High DBI (2 or more)	161 (37.6%)
Inappropriate prescriptions (IP), mean (SD)		3.14 (2.27)
Number of IP	0 IP	43 (10.0%)
	1 or more IP	385 (90.0%)
	2 or more IP	322 (75.2%)
	3 or more IP	246 (57.5%)

**Table 4 ijerph-18-11310-t004:** Descriptive analysis at baseline and number of Inappropriate Prescriptions (IPs): 0 (n = 43, 10.0%), 0–1 (n = 385, 24.8%), ≥2 (n = 322, 75.2%), 0–1 (n = 182, 42.5%), and ≥3 (n = 246, 58.5%).

		Inappropriate Prescriptions								
N = 428		0	≥1	*p*	0–1	≥2	*p*	0–2	≥3	*p*
Baseline Demographic Data										
Age, mean (SD)		85.5 (7.2)	85.5 (7.7)	0.990	85.20 (8.5)	85.63 (7.4)	0.618	86.0 (7.8)	85.2 (7.6)	0.273
Gender	Men	11 (7.7%)	132 (92.3%)	0.251	29 (20.3%)	114 (79.7%)	0.128	57 (39.9%)	86 (60.1%)	0.430
	Women	32 (11.2%)	253 (88.8%)		77 (27.0%)	208 (73.0%)		125 (43.9%)	160 (56.1%)	
Origin	Home	25 (11.9%)	185 (88.1%)	0.209	49 (23.3%)	161 (76.7%)	0.500	81 (38.6%)	129 (61.4%)	0.105
	NH	18 (8.3%)	200 (91.7%)		57 (26.1%)	161 (73.9%)		101 (46.3%)	117 (53.7%)	
Baseline Clinical, Functional and Cognitive Data										
Medication management		44 (9.3%)	54 (14.0%)	0.391	14 (13.2%)	44 (13.7%)	0.905	26 (14.3%)	32 (13.0%)	0.703
BI, mean (SD)		60.7 (31.1)	48.7 (32.0)	0.020 *	53.21 (32.4)	48.85 (32.0)	0.226	50.5 (31.9)	49.5 (32.3)	0.755
BI (degree)	IB ≥ 95	8 (15.7%)	43 (84.3%)	0.013 *	15 (29.4%)	36 (70.6%)	0.192	22 (43.1%)	29 (56.9%)	0.695
	IB 90–65	16 (13.3%)	104 (86.7%)		32 (26.7%)	88 (73.3%)		52 (43.3%)	68 (56.7%)	
	IB 60–25	12 (9.3%)	117 (90.7%)		32 (24.8%)	97 (75.2%)		52 (40.6%)	76 (59.4%)	
	IB ≤ 20	7 (5.5%)	121 (94.5%)		27 (21.1%)	101 (78.9%)		7 (5.5%)	121 (94.5%)	
Cognitive Status (dementia)	No dementia	10 (8.9%)	102 (91.1%)	0.546	28 (25.0%)	84 (75.0%)	0.730	45 (40.2%)	67 (59.8%)	0.608
	Mild	2 (3.2%)	60 (96.8%)		11 (17.7%)	51 (82.3%)		26 (41.9%)	36 (58.1%)	
	Moderate ^†^	19 (17.0%)	93 (83.0%)		32 (28.6%)	80 (71.4%)		50 (44.6%)	62 (55.4%)	
	Advanced ^‡^	12 (8.5%)	130 (91.5%)		35 (24.6%)	107 (75.4%)		61 (43.0%)	81 (57.0%)	
GS, mean (SD)		2.7 (1.5)	2.9 (1.5)	0.400	2,6 (1,4)	3.0 (1.5)	0.020 *	2.7 (1.4)	3.1 (1.5)	0.004 *
GS Type	Fall	10 (23.3%)	134 (34.8%)	0.128	27 (25.5%)	117 (36.3%)	0.040 *	54 (29.7%)	90 (36.6%)	0.134
	Dysphagia	5 (11.6%)	79 (20.5%)	0.164	16 (15.1%)	68 (21.1%)	0.176	29 (15.9%)	55 (22.4%)	0.098 *
	Pain	13 (30.2%)	86 (22.3%)	0.244	24 (22.6%)	75 (23.3%)	0.890	36 (19.8%)	63 (25.6%)	0.157
	Depressive Syndrome	16 (37.2%)	182 (47.3%)	0.209	41 (38.7%)	157 (48.8%)	0.071	68 (37.4%)	130 (52.8%)	0.001 *
	Insomnia	24 (55.8%)	205 (53.2%)	0.749	51 (48.1%)	178 (55.3%)	0.199	89 (48.9%)	140 (56.9%)	0.101
Morbidities, mean (SD)		4.1 (2.1)	5.0 (2.1)	0.014 *	4.2 (1.9)	5.1 (2.2)	<0.001 *	4.4 (2.0)	5.3 (2.2)	<0.001 *
Morbidities (number)	1–2	8 (18.6%)	35 (81.4%)	0.011 *	19 (44.2%)	24 (55.8%)	0.001 *	26 (60.5%)	17 (39.5%)	<0.001 *
	3–4	20 (11.9%)	148 (88.1%)		47 (28.0%)	121 (72.0%)		83 (49.4%)	85 (50.6%)	
	5 or more	15 (6.9%)	202 (93.1%)		40 (18.4%)	177 (81.6%)		73 (33.6%)	144 (66.4%)	
Morbidities (type)	Hypertension	22 (51.1%)	268 (69.6%)	0.003 *	61 (57.5%)	229 (77.1%)	0.002 *	107 (58.8%)	183 (74.3%)	<0.001 *
	Chronic renal failure	15 (34.9%)	171 (44.4%)	0.232	40 (37.7%)	146 (45.3%)	0.171	71 (39.0%)	115 (46.7%)	0.110
	Type 2 Diabetes	2 (4.7%)	84 (21.8%)	0.008 *	15 (14.2%)	95 (29.5%)	0.009 *	33 (18.1%)	77 (31.3%)	0.010 *
	Heart failure	6 (14.0%)	82 (21.3%)	0.258	17 (16.0%)	71 (22.0%)	0.184	34 (18.7%)	54 (22.0%)	0.408
FI, mean (SD)		0.34 (0.1)	0.39 (0.1)	0.023 *	0.36 (0.12)	0.40 (0.13)	0.013 *	0.36 (0.1)	0.40 (0.1)	0.007 *
FI (degree)	No FI	3 (9.4%)	29 (90.6%)	0.017 *	10 (31.3%)	22 (68.8%)	0.004 *	15 (46.9%)	17 (53.1%)	0.004 *
	Mild FI	17 (15.0%)	96 (85.0%)		33 (29.2%)	80 (70.8%)		52 (46.0%)	61 (54.0%)	
	Moderate FI	22 (10.9%)	179 (89.1%)		55 (27.4%)	146 (72.6%)		98 (48.8%)	103 (51.2%)	
	Severe FI	1 (1.2%)	81 (98.8%)		8 (9.8%)	74 (90.2%)		17 (20.7%)	65 (79.3%)	
End-of-life patients		11 (25.6%)	144 (37.4%)	0.126	34 (32.1%)	121 (37.6%)	0.307	64 (35.2%)	91 (37.0%)	0.697
Therapeutic Aim	Survival	3 (7.3%)	38 (92.7%)	0.198	11 (26.8%)	30 (73.2%)	0.702	18 (43.9%)	23 (56.1%)	0.754
	Functionality	28 (12.6%)	195 (87.4%)		58 (26.0%)	165 (74.0%)		98 (43.9%)	125 (56.1%)	
	Symptomatic	12 (7.3%)	152 (92.7%)		37 (22.6%)	127 (77.4%)		66 (40.2%)	98 (59.8%)	

* Statistically Significant. ^†^ Global Deterioration Scale (GSD) from 5 to 6B. ^‡^ Global Deterioration Scale (GSD) from 6C. Abbreviations: NH, Nursing Home; MD, mild dependence; BI, Barthel Index; GS, Geriatric Syndromes; FI, Frailty Index.

**Table 5 ijerph-18-11310-t005:** Inappropriate prescriptions identified considering the ATC (Anatomical, Therapeutic, and Chemical) classification system.

ATC Group	Total
A–Alimentary tract and metabolism	330 (24.7%)
B–Blood and blood-forming organs	124 (9.3%)
C–Cardiovascular system	412 (30.8%)
D–Dermatological	0
G–Genitourinary system and hormones	26 (1.9%)
H–Systemic hormonal preparations (excluding sex hormones and insulin)	7 (0.5%)
J–Anti-infective for systemic use	1 (0.1%)
L–Antineoplastic and immunomodulation agents	2 (0.2%)
M–Musculoskeletal system	28 (2.3%)
N–Nervous system	329 (24.6%)
R–Respiratory system	69 (5.1%)
S–Sensory organs	6 (0.4%)
V–Various	0

**Table 6 ijerph-18-11310-t006:** Univariate and multivariate analysis.

Patient Characteristics		Inappropriate Prescriptions					
		0 vs. ≥1		0–1 vs. ≥2		0–2 vs. ≥3	
OR		Univariate	Multivariate	Univariate	Multivariate	Univariate	Multivariate
Barthel Index (BI), mean (SD)		0.99 (0.98–0.99)					
BI (degree)	Indep.: ≥95	1	-	-	-	-	-
	Mild: 90–65	1.20 (0.48–3.03)					
	Mod.: 60–25	1.81 (0.69–4.74)					
	Severe: ≤20	3.2 (1.10–9.40)					
Geriatric Syndrome (GS), mean (SD)				1.19 (1.03–1.39)		1.21 (1.06–1.37)	
GS (degree)	0	-	-	1	-	1	-
	1–2			1.69 (0.6–4.6)		0.99 (0.37–2.62)	
	≥3			2.24 (0.8–6.0)		1.69 (0.65–4.41)	
Fall	Not	-	-	1	-	-	-
	Yes			1.67 (1.0–2.7)			
Depressive Syndrome	Not	-	-	-		1	-
	Yes					1.88 (1.27–2.78)	
T2DM	Not	1	-	1	-	1	-
	Yes	5.7 (1.3–24.1)		2.3 (1.2–4.5)		1.9 (1.2–3.2)	
Morbidities, mean (SD)		1.24 (1.04–1.48)		1.25 (1.11–1.41)		1.24 (1.12–1.37)	
Morbidities (number)	1–2	1	-	1	-	1	-
	3–4	1.69 (0.68–4.15)		2.04 (1.0–4.0)		1.57 (0.79–3.10)	
	≥5	3.08 (1.21–7.80)		3.5 (1.8–7.0)		3.02 (1.54–5.91)	
Frailty Index (FI), mean (SD)		15.09 (1.43–159.6)		8.27 (1.5–44.5)		7.63 (1.70–34.23)	
FI (degree)	None: 0–0.19	1	-	1	1	1	1
	Mild: 0.20–0.35	0.58 (0.16–2.13)		1.10 (0.5–2.5)	0.86 (0.34–2.12)	1.04 (0.47–2.27)	0.80 (0.54–1.89)
	Mod.: 0.36–0.50	0.84 (0.53–2.99)		1,21 (0.7–2.7)	0.97 (0.41–2.3)	0.93 (0.44–1.96)	0.82 (0.62–1.84)
	Severe: 0.51–1	8.38 (0.93–83.79)		4.21 (1.5–11.9)	2.62 (0.87–7.86)	3.37 (1.41–8.10)	2.13 (0.96–5.50)
Polypharmacy, mean (SD)		1.21 (1.09–1.34)	-	1.26 (1.17–1.36)	-	1.26 (1.19–1.35)	-
Medications (number)	0–4	1	1	1	1	1	1
	5–9	2.80 (1.36–5.77)	2.80 (1.36–5.77)	3.3 (1.9–5.6)	2.9 (1.7–5.1)	3.52 (1.99–6.22)	3.17 (1.77–5.68)
	≥10	4.55 (1.87–11.11)	4.55 (1.87–11.11)	6.9 (3.6–13.4)	6.07 (3.1–11.8)	9.75 (5.17–18.38)	8.65 (4.53–16.51)
MRCI, mean (SD)		1.03 (1.01–1.06)	-	1.05 (1.3–1.07)	-	1.05 (1.04–1.07)	-
MRCI (degree)	Low: 0–19,99	1	-	1	-	1	-
	Mod.: 20–39.99	3.6 (1.7–7.3)		2.76 (1.68–4.54)		2.80 (1.73–4.53)	
	High: ≥40	3.6 (1.4–8.8)		6.87 (3.31–14.24))		7.11 (3.87–13.06)	
DBI, mean (SD)		-	-	1.442 (1.08–1.91)	-	1.41 (1.11–1.79)	-
DBI (degree)	Low: 0–0.99	-	-	1	-	1	-
	Mod.: 1–1.99			1.49 (0.92–2.40)		1.25 (0.82–1.90)	
	High: ≥2			2.29 (1.13–4.68)		2.28 (1.27–4.11)	

Abbreviations: Indep., independence; Mod., moderate; OR, Odds ratio; MRCI, Medication Regimen Complexity Index; DBI, Drug Burden Index.

## Data Availability

The datasets generated during and/or analysed during the current study are available from the corresponding author on reasonable request.

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
