# Peer review of "Factors Associated with the Detection of Inappropriate Prescriptions in Older People: A Prospective Cohort"

_ijerph, 2021, doi:10.3390/ijerph182111310_

Round 1

Reviewer 1 Report

1.- Is different to talk about Community - Dwelling and to talk about of Nursing Homes.

2.- So, the title, it isn´t correct, because in the content, you are talking about of two different kind of Older Persons.

3.- I think your research, could be : "A comparison ...............between Older People at Community- Dwelling versus Older People at Nursing Homes"

4.- it could be interesting, that you talk about, The COVID 19, because at the beginning, the Medical Doctors didn´t know the correct therapy, some medicines were being used and then with the development of the research, some of them were withdrawals. (The date of your study included stage of pandemic).

5.- I advise you, to read this bibliography:

a.- Sofia Burato, Luca Leonardi, Ippazio Cosimo Antonazzo, Emanuel Raschi et Al.

"Comparing the prevalence of polypharmacy and potencial Drug -Drug Interactions in Nursing Homes and in the Community Dwelling  Elderly of Emilia Romagna Region".                     Front Pharmacol 11 February 2021.

In line with different rates of poly pharmacy 80% vs 16%.            The risk of exposure to at least one interaction was 53.7% in Nursing Homes and 26.4% in outpatients.

b.- Damien Cateau, Anne Niquille.

"Evolution of potentially inappropriate medication use in Nursing Homes: Retrospective analysis of drug consumption data".

Research in Social and Administrative Pharmacy.

Volume 17 Issue 4, April 2021 Pages 701 - 706.

c.- A Fernandez, F Gomez, CL. Curcio, E Pineda et Al.

"Prevalence and impact of potentially inappropriate medication on Community - Dwelling"        Biomed 2021, Vol 41, n 1 pp 111 - 122.

d.- Giada Ida Greco, MD, Marianna Noale, MSc and Giuseppe Sergi M.D PhD.

"Increase in Frailty in Nursing Home Survivors of Coronavirus Disease 2019. Comparison with Noninfected Residents".                                    

Journal of the American Medical Directors Association. 

2021 May; 22 (5): 943 - 947

Reviewer 2 Report

Important area highlighting the relationship between inappropriate polypharmacy, frailty and multimorbidity. Well written with only minor queries/comments which I hope the authors can address

  • What is ionogram?
  • The Patient Centered Prescription model is not one that is widely familiar to many people. Could the authors expand on it and perhaps put it in context with some of the other more familiar medication review models out there? 
  • What proportion is the 428 participants of the CEP cohort?
  • How were the Geriatric Syndromes selected for Table 2 and later in Table 4 chosen?
  • I found Figure 2 difficult to interpret. It was hard to appreciate the % visually. PIM was abbreviated and would benefit from writing it in full somewhere. Plus the line ≥1 includes both ≥2 and ≥3 which makes it all a bit redundant. A suggestion only, but would a bar chart (or stacked bar chart) be better?
  • Frailty through out the manuscript was presented categorically but then analysed as continuous here: Patients with one or more IP presented a higher mean of FI than 245 those without IP. 
  • Could the authors justify why the need to do the multiple different group comparisons, 0 vs ≥1, 0-1 vs ≥2, 0-2 vs ≥3? 
  • Lastly, were there differences in participant characteristics, variables and outcomes between those living in care home and non-care home setting?

Thank you
